# Bacterial Infections and Cancer: Exploring This Association And Its Implications for Cancer Patients

**DOI:** 10.3390/ijms24043110

**Published:** 2023-02-04

**Authors:** Kafayat Yusuf, Venkatesh Sampath, Shahid Umar

**Affiliations:** 1Department of Surgery, University of Kansas Medical Center, Kansas City, KS 66160, USA; 2Department of Cancer Biology, University of Kansas Medical Center, Kansas City, KS 66160, USA; 3Department of Pediatrics and Gastroenterology, Children’s Mercy Hospital, Kansas City, KS 66160, USA

**Keywords:** bacterial infections, cancer, antibiotics, antibiotic resistance, adaptation, dormancy, microbiome, dysbiosis, cancer patients

## Abstract

Bacterial infections are common in the etiology of human diseases owing to the ubiquity of bacteria. Such infections promote the development of periodontal disease, bacterial pneumonia, typhoid, acute gastroenteritis, and diarrhea in susceptible hosts. These diseases may be resolved using antibiotics/antimicrobial therapy in some hosts. However, other hosts may be unable to eliminate the bacteria, allowing them to persist for long durations and significantly increasing the carrier's risk of developing cancer over time. Indeed, infectious pathogens are modifiable cancer risk factors, and through this comprehensive review, we highlight the complex relationship between bacterial infections and the development of several cancer types. For this review, searches were performed on the PubMed, Embase, and Web of Science databases encompassing the entirety of 2022. Based on our investigation, we found several critical associations, of which some are causative: *Porphyromonas gingivalis* and *Fusobacterium nucleatum* are associated with periodontal disease, *Salmonella* spp., *Clostridium perfringens*, *Escherichia coli*, *Campylobacter* spp., and *Shigella* are associated with gastroenteritis. *Helicobacter pylori* infection is implicated in the etiology of gastric cancer, and persistent *Chlamydia* infections present a risk factor for the development of cervical carcinoma, especially in patients with the human papillomavirus (HPV) coinfection. *Salmonella typhi* infections are linked with gallbladder cancer, and *Chlamydia pneumoniae* infection is implicated in lung cancer, etc. This knowledge helps identify the adaptation strategies used by bacteria to evade antibiotic/antimicrobial therapy. The article also sheds light on the role of antibiotics in cancer treatment, the consequences of their use, and strategies for limiting antibiotic resistance. Finally, the dual role of bacteria in cancer development as well as in cancer therapy is briefly discussed, as this is an area that may help to facilitate the development of novel microbe-based therapeutics as a means of securing improved outcomes.

## 1. Introduction

Cancer is a disabling, challenging, and frightening disease that can affect any body part [1]. The global cancer epidemic remains a public health concern, as cancer is established as the second foremost cause of death in the United States, with close to 2 million new cases and a little over 600,000 deaths expected to occur in 2022 [2]. Mortality from lung cancer remains the most common type of cancer-related death, accounting for approximately 350 deaths daily [2]. The public health burden associated with these high numbers has fueled massive research efforts to uncover the preventable causes of cancer [3]. In a series of excellent reviews published recently, expert analysts reflect on the fact that the global burden of cancer is growing, especially in the most vulnerable sociodemographic populations, and that significant global effort is required not only to reduce the incidence of cancer but also to provide more equally distributed cancer control [4,5,6].

Cancer results from a series of genetic and epigenetic changes that disrupt regular cell growth, control, and survival. A wide range of intrinsic and extrinsic factors influence these changes. Intrinsic factors may include genetic mutations, random errors in DNA replication, immune and inflammatory responses along with other modifiable factors, including aging; external factors may include diet type, smoking/tobacco use, radiation, and infectious organisms [7].

Infectious pathogens such as bacteria and viruses are modifiable causes of cancer, accounting for 20% of all human tumors [8,9]. Pathogens associated with cancer can exhibit mechanisms that include persistent infection, evasion of the immune response, chronic inflammation leading to continued cell proliferation, and an increased risk of oncogenic transformation, even in immune-competent individuals [10].

The human body is home to many microbes that form complex ecological habitats and influence the physiology of human health and disease, the totality of which may be summarized as the human microbiome [11,12]. The most effective way to describe the human microbiome is as a complex collection of microorganisms found in different body parts, including the skin, the oral cavity and saliva, the respiratory system, the reproductive tract, and the gastrointestinal system. These microorganisms include bacteria, eukaryotes, archaea, fungi, and viruses [12,13]. Since the population of bacteria in the microbiome vastly outnumbers that of other microorganisms, researchers sometimes simply refer to the microbiome as bacteria [13]. According to Curtis and Sperandio, there are 100 trillion bacteria in the human body, with 500–1000 different bacterial species living mostly (but not exclusively) in the gut [14]. The resulting interactions are beneficial to our homeostasis and well-being [15].

Commensal bacteria colonize the host shortly after birth, forming at first a small community that progressively transforms into a diversified ecosystem. Over time, the host-bacterial associations develop into a mutually beneficial relationship [14,16]. The gut, for instance, provides nutrients to resident bacteria, which in turn assists food digestion, the absorption of nutrients, and the metabolism of indigestible substrates. This coexistence can also help to modulate immune system activity, maintain the intestinal architecture, and prevent the colonization of pathogenic microorganisms [14,16]. Despite this, an imbalance in host-bacterial interactions (dysbiosis) can change the physiological equilibrium in the host cells. This discrepancy can facilitate the progression of many conditions including inflammatory bowel disease, malnutrition, obesity, diabetes, and cancer [15,17].

Given the intrinsic relationship between humans and bacteria, specific bacterial pathogens responsible for cancer morbidity, mortality, and treatment resistance must be highlighted to help identify new therapeutic approaches.

## 2. Methods

In preparation for this review, articles describing bacterial infections and cancer were retrieved using these search terms: Microbes (MeSH Terms), Bacteria (MeSH Terms, AND Cancer (MeSH Terms)). Articles related to the use of antibiotics, antibiotic resistance, or adaptation, etc., were retrieved using these search terms: Antibiotics (MeSH Terms) OR Resistance (MeSH Terms) OR Adaptation (MeSH Terms). All these searches were performed on the PubMed, Embase, and Web of Science databases and care was taken to include all of the recent discoveries relating to this topic. The results were further screened by title and abstract. Articles not directly relevant were excluded. Overall, the intent was to provide readers with a review of cutting-edge science on the role of bacterial infections in cancer in general and the ways in which some of these infections might stimulate inflammatory signals to promote neoplastic transformation.

## 3. The Link between Bacterial Infection and the Onset of Cancer

Bacteria and bacterial infections can act as potential carcinogens and tumor promoters [18]. The production of bacterial toxins, enzymes, and oncogenic peptides can all significantly contribute to tumor development by promoting inflammation, interfering with cell cycle control, and disrupting cell signaling pathways [18,19]. Other studies have also corroborated the fact that microbiota-mediated infection stimulates cancer cell proliferation by targeting host cell DNA, altering the immune system, and promoting epithelial-to-mesenchymal transition [17]. A description of the most frequently studied cases highlighting the link between bacterial infections and cancer is summarized in Figure 1.

### 3.1. Helicobacter pylori Infection and Gastric Cancer

One of the most well-known and researched connections associating cancer with a bacterial origin is the link between *Helicobacter pylori* infection and gastric cancer [8,20,21]. *H. pylori* has been extensively studied in this context with comprehensive epidemiological data supporting the existence of its causal relationship with carcinogenesis [20,22]. *H. pylori* is a gram-negative spiral-shaped bacterium that lives in a neutral pH niche between the stomach mucus layer and the gastric epithelium [23,24,25]. Studies have opined that there are two potential explanations for the association between *H. pylori* and gastric cancer. The first of these is that *H. pylori* infection causes persistent gastric mucosal inflammation, resulting in atrophy and eventual intestinal metaplasia; the second is that *H. pylori* can create, alter, or release bacterial virulence factors that play a significant role in cancer progression [20,23,25]. Studies have also shown that *H. pylori* infection contributes to an increase in the bacteria populations from the Proteobacteria, Spirochaetes, and Acidobacteria phyla, alongside a decline in the abundance of Bacteroidetes, Actinobacteria, and Firmicutes [26]. *H. pylori* infection promotes persistent inflammation of the gastric mucosa and facilitates the progression of both mucosal-associated lymphoid tissue lymphoma and gastric cancer [27,28].

Causes and incidence: *Helicobacter pylori* is a ubiquitous bacterium with a long history. Its colonization rates vary widely, with the infection rates in developing countries being higher than those in developed regions [29]. Most infections are reported as being contracted in childhood through the feces-to-mouth or the mouth-to-mouth routes [30]. Different outcomes of *H. pylori* infection may also depend on the specific strain of the infection and its bacterial properties, the host’s response to the infection, and environmental factors; all of these influence pathogen–host interactions [31].

Adaptation strategies for survival and the evasion of treatment: *H. pylori* has proven its ability to infiltrate the stomach and survive there, even though the stomach is an acidic environment to many species. It is almost impossible for the host to completely eliminate the infection, as *H. pylori* uses locomotion, chemotaxis, urease synthesis (Figure 1), and other strategies to adapt to the harsh stomach environment and to persist for decades [32]. Furthermore, *H. pylori* secrete VacA, a vacuolating pore-forming cytotoxin [33]. VacA pores can disrupt mitochondria and endosomes, modify plasma membrane permeability, and cause leakage in epithelial monolayers [33]. VacA also promotes apoptosis in many cells by interfering with mitochondrial activity. In addition to being cytotoxic, VacA can suppress the immune system and prevent the proliferation of T-cells [32,33]. Although all *H. pylori* strains possess the VacA gene [34], certain species of the bacteria have mutations in their VacA sequences, and patients infected with strains harboring the s1, i1, or m1 variants of VacA have an increased risk of developing gastric cancer [34].

### 3.2. Periodontal Disease Caused by Bacterial Infections Can Promote Tumorigenesis

Periodontal disease is a common oral bacterial infection that causes inflammation of the gums (gingivitis) and the tooth-supporting structures (periodontitis) [35]. Periodontal disease can promote long-term and persistent inflammation that can eventually lead to systemic diseases such as cancer of the esophagus [36]. Several types of bacteria have been identified as periodontal pathogens; the most extensively studied are *Porphyromonas gingivalis* and *Fusobacterium nucleatum*, which have been shown to stimulate oral tumor proliferation [37]. These extensively studied organisms in the gums have become the focus of the emerging link between oral bacteria and cancer [38].

*Fusobacterium nucleatum* is an anaerobic, gram-negative, disease-causing bacterium in the oral cavity [39]. Studies have suggested that dysbiosis of the oral microbiome allows *F. nucleatum* to become an opportunistic bacterium that can cause gum disease and human cancers [38,39]. A research study revealed that the abundance of *F. nucleatum* in the tissues of patients with esophageal squamous carcinoma was linked to shorter patient survival times [39].

*Porphyromonas gingivalis* is a gram-negative bacterium that is an apparent pathogen in periodontal diseases and other systemic conditions [40]. In 2011, a detailed study found that the concentration of *P. gingivalis* was higher in cancer cells than in normal mouth tissues. The authors suggested that *P. gingivalis* promoted oral carcinogenesis by transforming epithelial cells [41]. Correspondingly, a meta-analysis discovered that *P. gingivalis* significantly increased the risk of periodontal disease and cancer [42].

According to recent research, pathogenic bacteria that cause periodontitis may actively enter the bloodstream from periodontal tissues, become dislodged and carried to numerous organs, and increase the risk of inflammatory processes in those organs [43]. In addition to this, some researchers have found promising evidence linking periodontitis-related bacteria to other cancers [38,44]. According to the reports, *F. nucleatum* is prevalent in colorectal adenomas and advanced-stage colorectal cancer (CRC) [45]. It was discovered that *F. nucleatum* causes an inflammatory response that enables the persistent proliferation of colorectal cancer cells [45]. In a similar study, in comparison with healthy volunteers, patients with pancreatic cancer produced more elevated levels of antibodies against *P. gingivalis*. The study revealed that patients with more elevated levels of *P. gingivalis* antibodies had a twofold higher risk of developing pancreatic cancer [36]. Investigators from the Harvard School of Public Health conducted a more recent study analyzing the link between periodontal disease, tooth loss, and the risk of gastric and esophageal cancer in approximately 98,000 women (1992 to 2014) and approximately 49,000 men from the Health Professionals Follow-Up Study (1988 to 2016). According to the findings, the risk of gastric cancer was elevated by 52% and the risk of esophageal cancer by 43% in people with periodontal disease [46].

Causes and incidence: Periodontal disease is a well-established and common oral condition in the human population [47]. Periodontal diseases are associated with several risk factors: smoking habits, poor oral hygiene, medication regimens, age, heredity, and stress [47]. These factors can hasten the dysbiosis of the oral microbiome and the multiplication of pathogenic bacteria, which drive disease progression.

Adaptation strategies for survival and the evasion of treatment: One major mechanism related to the persistence and adaptability observed in oral pathogenic bacteria is their ability to form dental plaques or biofilms (Figure 1). These structures make them resistant to mechanical stress or antibiotic treatment [47,48]. The plaque biofilm acts as a protective barrier by isolating them from harmful agents and maintaining distinct phenotypic characteristics that promote their survival. Once the niche is established, the bacteria can release toxins such as lipopolysaccharide (LPS) and produce proinflammatory cytokines that promote chronic inflammation in periodontal tissues [49].

### 3.3. Bacterial Pathogenesis in the Onset of Gastroenteritis, Acute Diarrhea, and Colon Cancer

Bacterial gastroenteritis is a kind of inflammation in the stomach and small intestine caused by a bacterial infection. In affected patients, the disease is often accompanied by severe diarrhea [50]. Although gastroenteritis can be caused by viruses, fungi, or parasites, most cases are caused by bacterial pathogens [50]. Bacteria associated with gastroenteritis include *Escherichia coli*, *Salmonella* spp., *Clostridium perfringens*, *Campylobacter* spp., and *Shigella* [50]. *Salmonella* and *E. coli* are the most frequently studied regarding their potential role in carcinogenesis.

*Salmonella* is an intracellular infection that affects a variety of animals and humans. The results of a *Salmonella* infection can range significantly from a moderate, self-limiting gastroenteritis to a severe and potentially fatal systemic infection [51]. *Salmonella enterica* serovar Enteritidis has been implicated in several gastroenteritis outbreaks linked to contaminated food [52,53]. An exhaustive study found that episodes of bacterial gastroenteritis caused by *Salmonella* or *Campylobacter* infection resulted in an increased risk of inflammatory bowel disease only one year after acute bacterial gastroenteritis [54].

A more recent epidemiological investigation supported these findings when it discovered a link between severe *Salmonella* infection and an elevated risk of colon cancer [55]. According to a further study, *Salmonella* encourages the development of colonic tumors; its AvrA protein can stimulate the Wnt and STAT3 signaling pathways in colonic tumor cells [56].

*E. coli* is a gram-negative bacterium, ubiquitous in nature, that is found in the human gut microbiome [57]. Several studies have found that patients with colorectal cancer have higher levels of colon mucosa-associated *E. coli* colonization than those found in healthy individuals [57,58]. Pathogenic *E. coli* strains have been shown to produce toxins (Figure 1) such as cyclomodulin, cytotoxic necrosis factor (CNF), circulation inhibitory factor (Cif), colibactin, and cytolethal distending toxins (CDT). By disrupting the cell cycle and/or promoting DNA damage, these toxins can affect cell differentiation, apoptosis, and cell proliferation [57,59,60].

Enterotoxigenic *Bacteroides fragilis* (ETBF) is another common bacterium associated with acute diarrhea infections [61]. *Bacteroides fragilis* (*B. fragilis*) is an obligate anaerobic gram-negative bacillus bacterium [62] that makes up 0.1% of the normal flora of the colon [25]. However, ETBF levels are elevated in CRC patients’ feces and colonic mucosal tissues [25]. According to the “alpha-bug” theory, a critical pathogenic species, such as (ETBF), remodels the microbiota to promote CRC, most probably through an IL-17 and Th17-mediated inflammatory response [63]. Similarly, the bacterial driver–passenger view proposes that “driver” bacteria, such as ETBF, impel or exacerbate inflammation while producing toxins that cause cell proliferation and mutations. Consequently, an adenoma develops and is colonized by “passenger” bacteria, such as *Fusobacterium* spp., thus promoting tumor growth [64].

Causes and incidence: Acute gastroenteritis affects millions of people worldwide and is characterized by symptoms such as vomiting, diarrhea, and abdominal pain [65]. Acute gastroenteritis caused by bacterial pathogens is acquired through contaminated foods. Researchers also believe that the widespread use of antibiotics [66] in recent years has aided pathogenic bacterial colonization [51].

Adaptation strategies for survival and the evasion of treatment: Pathogenic bacteria have evolved adaptations to their outer surfaces that permit immunological escape and opportunities to survive and withstand the host’s defense mechanisms against infection [8]. Gram-negative bacteria such as *E. coli* encapsulate their intricate outer-surface macromolecules in a polysaccharide-rich capsule to prevent immune-system detection and clearance. These capsules are protective and help minimize the host’s complement activation [8,67]. The release of protein toxins with cytolytic characteristics is a further adaptive techniques used by pathogenic bacteria [8]. *Shigella dysenteriae*, *Campylobacter jejuni*, *Salmonella typhi*, and *E. coli* all secrete cytolethal distending toxins (CDTs); these toxins promote DNA strand breaks that eventually promote cancer progression [8,68]. In addition, *B. fragilis* releases a toxin (BFT, Figure 1) that attaches to gut epithelial cell receptors and promotes cell proliferation by cleavage of E-cadherin [62,69].

### 3.4. Chlamydia pneumoniae Infection and Lung Cancer

*Chlamydia pneumoniae* (Cpn) is an obligate intracellular bacterium that causes various respiratory diseases in humans, including pneumonia [70]. Chronic obstructive pulmonary disease, asthma, and lung cancer may result from repeated or prolonged exposure to *Chlamydia* antigens [71].

Previous research has found serological evidence of a link between *Chlamydia pneumoniae* infection and lung cancer [72]. A report also reviewed previous epidemiological studies on the link between *C. pneumoniae* and lung cancer. It concluded that earlier investigations supported a causative relationship between *C. pneumoniae* infection and lung cancer [73]. Correspondingly, a group of researchers conducted a meta-analysis on 12 published articles that studied the link between *C. pneumoniae* infection and lung cancer. The study concluded that *C. pneumoniae* infection is correlated with an elevated risk of lung cancer, implying that a higher serological titer may be an efficient predictor of lung cancer risk [74]. A systematic review of published articles was recently conducted on *C. pneumoniae* infection and the development of lung cancer. The authors of this study discovered that twenty-four articles describing one animal model positively supported the hypothesis, while three of the articles did not [70].

More exhaustive studies on different animal models and follow-up studies in humans with Cpn are urgently needed to assist in validating these findings.

Causes and incidence: Cpn, a gram-negative bacillus, is an obligate intracellular bacterium responsible for respiratory infections [75,76]. Cpn is aerosolized by individuals coughing or sneezing, which produces tiny respiratory droplets that contain bacteria. Others then inhale these droplets and the bacteria that cause the infection [75].

Adaptation strategies for survival and the evasion of treatment: *Chlamydia*’s most common adaptative strategy is to enter a persistent dormant state (Figure 1). If *Chlamydia* is exposed to immune system stress or antibiotics, the pathogen can suspend replication by entering a dormant but viable state called “*Chlamydia* persistence”. During this time, *Chlamydia* prioritizes the cellular activities required for extended survival; this is often associated with the presence of enlarged abnormal reticulate bodies. When conditions improve and normal replication resumes, the durable state is reversed [77].

### 3.5. Salmonella typhi Infections and Gallbladder Cancer

*Salmonella typhi* is a gram-negative, rod-shaped, flagellated bacterium that is well documented as being responsible for typhoid infections [78,79,80]. After infection, *S. typhi* invades the gallbladder, causing persistent infection in susceptible carriers that serve as a reservoir for the spread of the disease; this persistent chronic infection has also been linked to cancer [79,81]. The ability of the pathogen to survive and persist for an extended time in the gallbladders of patients after infection creates a suitable environment for its tumor-promoting effect [81]. It has also been reported that *S. typhi* produces typhoid toxins and carcinogenic toxins such as nitroso-chemical compounds. These compounds play a crucial role in the progression of cancer [80]. A long-term follow-up study of infected mice discovered the presence of precancerous lesions and dysplasia of the gallbladder, reflecting an association between *Salmonella* infection and gallbladder cancer [82]. Other large-scale studies have also found a link between *S. typhi* infection and gallbladder cancer [80,81,83].

Causes and incidence: Infection by *S. typhi* is usually acquired through the ingestion of food or water contaminated with the feces of a person carrying the organism [78]. *Salmonella typhi* is suspected to be responsible for approximately 22 million cases of typhoid fever, slightly more than 5 million cases of paratyphoid fever, and approximately 200,000 deaths worldwide each year [79]. According to research, some 90% of chronic infection carriers suffer from gallstone infections over time. This association has been described as a significant risk factor for developing gallbladder cancer [82,84,85].

Adaptation strategies for survival and the evasion of treatment: Through the secretion of biofilm, *S. typhi* appears to survive and become well adapted to thrive in the gallbladder. Biofilm formation (Figure 1) is a pathological adaptive response that allows the bacteria to aggregate, adhere to surfaces, and develop antimicrobial resistance through the utilization of a thick protective extracellular matrix [79,85].

### 3.6. Bacterial Infection and Cervical Cancer

*Chlamydia trachomatis* is a gram-negative bacterium and a prevalent cause of curable sexually-transmitted bacterial infections (STIs) worldwide [86]. The infection presents as urethritis in men and endocervicitis in women [87]. Persistent *Chlamydia* infections have also been identified as a risk factor for developing cervical carcinoma [87], especially in patients with the human papillomavirus (HPV) coinfection [86,88]. Recent elaborate meta-analysis studies have also corroborated the link between *C. trachomatis* and cervical cancer [89]. The mechanism of the *C. trachomatis* action is proposed as resulting from the dysregulation of DNA damage responses, and the disruption of cell-cycle control, which promotes malignant transformation in infected patients [88].

The association between *C. trachomatis* and cervical cancer is usually seen in the presence of HPV infections [86]. Since HPV infections have long been implicated in the etiology of cervical cancer [90], it is difficult to directly classify the bacterial infection as solely causative, even though it contributes significantly to an increased incidence.

## 4. Mechanism of Cancer Development after Bacterial Infection

The onset of carcinogenesis after exposure to bacterial infection occurs over an extended period and depends on a wide range of factors such as host susceptibility, genetic background, and suitable environmental conditions [91].

After the successful infection of the host either through oral, airborne, or sexually-transmitted routes (Figure 2), the infection may start as mild, acute, or chronic and may be symptomatic or asymptomatic [92]. Antibiotics might be introduced at this point to help ease the infection [92]. The initial infection and subsequent antibiotic use may then initiate bacterial dysbiosis in the microbiome of the infected organ, in this case promoting the bloom of opportunistic and pathogenic bacteria species [93,94,95]. Antibiotic use may completely eradicate the infection in some cases, or it may be ineffective against the application of alternative survival strategies by the bacteria to facilitate its continued existence over extended periods of time (Figure 2). Pathogenic bacteria can survive and persist in the host environment through dormancy, secretion of cyotethal toxins, the formation of protective biofilms, or mutations to form antibiotic resistant strains [17,18]. This adaptive technique enables bacteria to promote persistent chronic inflammation or chronic infection, and ultimately carcinogenesis [17] (Figure 2).

## 5. Bacterial Infections during Cancer Treatment

Bacterial infection is one of the most common complications during cancer treatment [96]. Although cancer mortality rates have continued to decline in recent years, bacterial infections remain a significant cause of infection-related mortality in patients [97]. Cancer patients are at a high risk of bacterial infection due to surgical complications, chemotherapy and radiotherapy-related neutropenia, and the use of immunosuppressive drugs during cancer treatment [97,98].

Several reports highlight that bacterial infections are prevalent in patients with blood cancers [99,100,101], owing mainly to prolonged neutropenia during treatment [101]. Although patients with solid tumors are also susceptible to bacterial infections, an extensive epidemiological study reported an eightfold higher incidence in patients with hematological cancers than in patients with solid tumors [102]. A more recent study also revealed that the levels of *Pseudomonas aeruginosa* bloodstream infections were higher in hematological malignancies than in solid tumors [103].

Extensive studies have identified a group of six bacterial microbes otherwise classified as the “ESKAPE” pathogens, which are at the forefront of bacterial infection and antibiotic resistance during cancer treatment (Figure 3) [104,105]. These organisms include *Acinetobacter baumannii*, *Staphylococcus aureus*, *Enterococcus faecalis*, *Klebsiella pneumoniae*, *Pseudomonas aeruginosa*, and *Enterobacter* spp. [104,105].

An extensive study conducted by a group of researchers at Cairo’s National Cancer Institute aimed to investigate the effect of ESKAPE pathogens on the course of infectious incidents in pediatric cancer patients. According to the study, the ESKAPE pathogens were significantly associated with more prolonged infections, longer duration of infection, and higher mortality. The study also found that *Pseudomonas aeruginosa* and *Klebsiella pneumoniae* infections caused the highest mortality rates, at 43% and 30%, respectively [106].

A similar rigorous study was carried out over a five-year period in hospitalized adults with cancer from 2006 to 2011. All bacteremia episodes in hospitalized cancer patients were included in the investigation. The study concluded that the ESKAPE pathogens were responsible for 34% of bacteremia episodes in cancer patients, with *Pseudomonas aeruginosa* being the most frequently isolated organism [107].

Most enterococcal infections in cancer patients are bacteremia caused by *Enterococcus faecalis* [108]. Prior exposure to antibiotics and long-term neutropenia are factors that predispose cancer patients to enterococcal infections [108]. Urinary tract infections (UTIs), bloodstream infections (BSIs), and endocarditis are common complications for cancer patients exposed to enterococcal infections during therapy [109,110]. Reports have also highlighted that *Enterococcus faecalis* often evades treatment owing to its ability to secrete virulent factors and biofilms, and the plasticity of its genome [111].

*Staphylococcus aureus* has a significant clinical impact on mortality rate in patients with malignancy [107]. A systematic analysis of bacteremia infections in cancer patients highlights that *Staphylococcus aureus* infections accounts for 1.3% to 12% of all bacteremia cases [112]. Infections caused by *Staphylococcus aureus* often reportedly led to BSIs, skin infections, pneumonia, and endocarditis [113]. *Staphylococcus aureus* is of clinical interest because the bacterium can form small colony variants (SCV) in addition to releasing virulence factors (e.g., enterotoxins and proteases) that interfere with the host’s innate immune response [114,115].

*Klebsiella pneumoniae* is a leading cause of sepsis and the most common cause of bacteremia, pneumonia, wound infections, abscesses, and urinary tract infections in cancer patients [116]. *K pneumoniae* causes a significant threat to health because the bacteria rely on the presence of an accessory genome that allows for genetic plasticity, the release of virulence factors, and the formation of a thick protective polysaccharide capsule that enables them to overcome the host immune system and clearance [117,118].

*A. baumannii* infections are nosocomial infections that can be fatal in patients with compromised immune systems, especially cancer patients. Patients infected with *Acinetobacter baumannii* have a significantly increased mortality rate of up to 80% [119]. Patients also develop complications from BSI, respiratory tract infections, meningitis, UTIs, and skin infections [113,120]. The pathogenesis of *A. baumannii* infection is severe because the bacterium can adapt to the host environment in several ways, which include the secretion of biofilms (thick polysaccharide capsules) and porins, alongside the release of enzymes and virulence factors. The bacterium also utilizes micronutrient acquisition and protein secretion systems that facilitate its survival even under harsh environmental conditions [120]. 

*Pseudomonas aeruginosa* is an increasingly prevalent opportunistic pathogen that can cause serious and life-threatening nosocomial infections [116]. *P. aeruginosa* has been implicated in causing BSI, respiratory infections, and endocarditis [113,121]. The bacterium is highly adaptive to the host environment owing to its robust genome and its ability to form biofilms, activate a motile-sessility switch, and secrete secondary metabolites [122].

*Enterobacter* spp. can also infect the respiratory tract, surgical wounds, the urinary tract, and the bloodstream of cancer patients [123]. These bacteria facilitate soft-tissue infections, BSIs, and endocarditis [113]. Often, the bacteria population thrives in its host environment through the use of flagellum, the formation of biofilms, and the secretion of endotoxins [124].

## 6. Antibiotic Use and Antimicrobial Resistance in Cancer Patients

Antibiotics are secondary metabolites produced by microorganisms, higher animals, and plants, and they have antipathogenic effects that can interfere with the growth of other living cells [125]. Antibiotics are increasingly being used to treat cancers, often because of their pro-apoptotic, antiproliferative, and antimetastatic potentials [125]. Since infections are also common in cancer patients, it is essential to use antibiotics to prevent and treat bacterial infections [98].

A major limitation of antibiotic use is the causation of dysbiosis due to the elimination of beneficial bacterial groups, such as Lactobacillus and Bifidobacterium, in addition to the pathogenic bacteria [125]. A further limitation is the ability of pathogenic bacteria to evade killing and induce antibiotic resistance [104].

It is also worth noting that the microbiome plays a crucial role in the development of antibiotic resistance [126,127]. Microbiota dysbiosis induced by antibiotic treatment contributes to the development of resistance that is often due to increased numbers of opportunistic bacteria secreting high levels of antimicrobial resistance genes [128,129].

Antibiotic resistance in cancer patients often correlates with increased susceptibility to infections that eventually reduce the patient’s survival time; it also poses a significant threat to the accomplishments achieved in cancer treatment, and it highlights the importance of monitoring cancer patients and protecting them against antibiotic resistance [104].

### Potential Approaches for Reducing Antibiotic Resistance in Cancer Patients

Avoiding bacterial infections (prevention is always better than cure): The first stage of bacterial-driven carcinogenesis is usually an initial infection that allows the pathogen to infect the host. Hosts can prevent such infections by being vaccinated and by adopting proper hygiene measures [130]. Antibiotic prophylaxis is another common practice used for cancer patients in neutropenic settings to prevent infection and infection-related complications. According to reports, the prophylactic use of quinolones reduced the incidence of fever, infection, hospitalization, and overall mortality [104].The discovery and design of novel antibiotics that are pathogen-specific: Most antibiotics are designed for broad-spectrum use and can severely impact the human microbiome, leading to complications such as antibiotic-associated colitis and the rapid spread of antibiotic resistance. Utilizing more targeted approaches or using narrow-spectrum antibiotics may help address this limitation and provide additional resources for expanding the antibiotic pool, which is at risk owing to the inevitable rise in resistance [131].Optimizing Antibiotic Use (more does not guarantee better): Excessive antibiotic use is a significant driver of resistance [132]. To promote the effective use of antibiotics, the concept of “antimicrobial stewardship” was developed. Antimicrobial stewardship refers to the practice of establishing the antibiotic treatment that is most appropriate and then administering it for a short time and at the lowest effective dose to achieve the best possible clinical outcome in combating infection and preventing its spread [133]. In clinical settings, antimicrobial stewardship teams comprise an infectious disease doctor, an infectious disease pharmacist, and clinical microbiologists who monitor the patient’s dosage and response [104]. A research study evaluated the association between patient mortality and antibiotic stewardship outcomes in patients who developed febrile neutropenia during their cancer treatment. The study showed that adherence to antibiotic stewardship was associated with reduced mortality [134].

## 7. Conclusions

The link between infectious agents and the risk of cancer has been studied for decades. Nevertheless, there are limited reports on the causal relationship between bacterial infection and tumorigenesis. The link between bacterial infections acquired during cancer treatment and mortality rates is yet to be explored. This comprehensive review highlights the association between bacterial infection and cancer growth, and it also demonstrates the impact of nosocomial bacterial infection on the survival of cancer patients. We also provide crucial information on how bacterial infections can persist for long durations, contribute to tumorigenesis, and affect cancer patients during their treatment. The article correspondingly provides recommendations for minimizing antibiotic resistance in cancer patients. To better understand the direct link between bacterial infections and cancer development, it will be essential to conduct further long-term follow-up studies on patients with bacterial infections and on the ways in which these infections impact on their risks. Finally, while we have focused in this review on the role of bacteria in cancer, recent studies have also shed light on bacteriotherapy as a potential anticancer therapeutic strategy. Live attenuated strains, toxins, peptides or bacteriocins could be used; alternatively, bacteria could be developed as a drug-delivery system for cancer therapy with the assistance of microbe engineering for tumor targeting. The therapeutic effects of engineered microorganisms in cancer treatment hold immense promise for biomedical applications in targeted cancer therapy. These areas, while cutting edge, are beyond the scope of this review and will be deferred for future discussion.

## Figures and Tables

**Figure 1 ijms-24-03110-f001:**
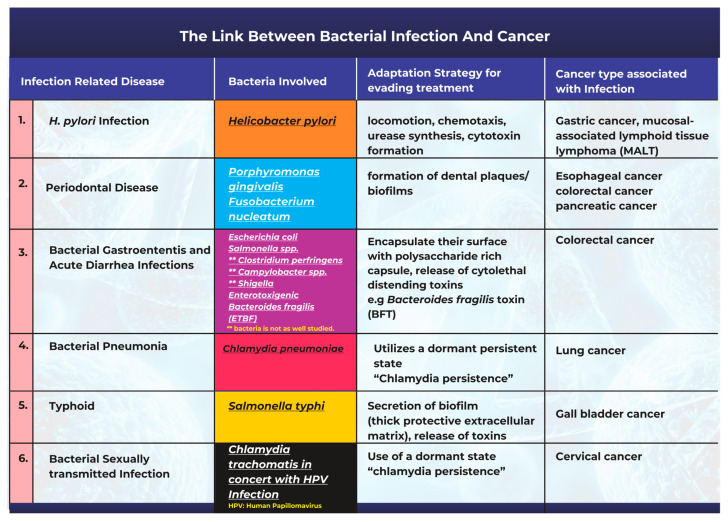
An illustration of the links between common bacterial infections and cancer.

**Figure 2 ijms-24-03110-f002:**
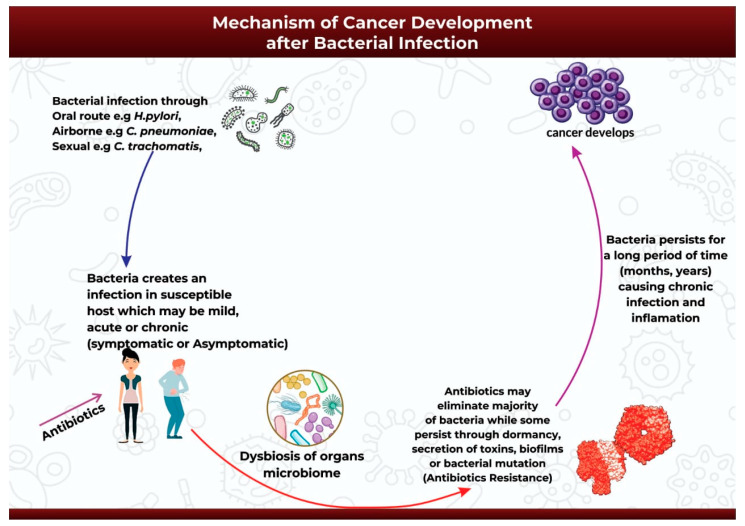
An illustration of the mechanism of cancer development after bacterial infection.

**Figure 3 ijms-24-03110-f003:**
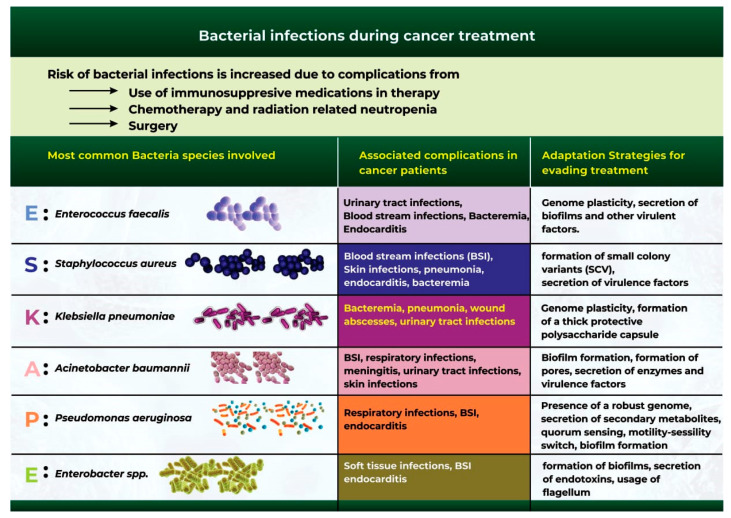
The most common bacterial species that cause bacterial infections during cancer therapy.

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
