# Peer review of "Bacterial Infections and Cancer: Exploring This Association And Its Implications for Cancer Patients"

_ijms, 2023, doi:10.3390/ijms24043110_

Round 1

Reviewer 1 Report

The manuscript entitled “Bacterial infections and cancer: exploring the association and its implications for cancer patients” by Kafayat Yusuf et al., is well written and well presented.

Authors highlights a very important issue of the association between bacterial infections and cancers.

Paper included very important research work published on the role of different bacterial infections and how they aid in cancer progression.

I have some suggestions as follows.

Rewrite abstract of the paper and write key results in the abstract. The length of abstract is also short. Increase it to 300 words. Add the linkage of major bacterial infections with cancer in the abstract.

The typical abstract is comprise of background, methods (search criteria, studies included etc), Results (for review it should be major part, with inclusion of key bacterial infections and their role in cancer), conclusion (2 sentences).

There are only a couple of papers published in 2022, which are included in the study. Add some latest data

Cite global cancer burden from the latest papers published in 2022 by Global Burden of Disease Research group, University of Washington, as pasted below.

1.     The global burden of cancer attributable to risk factors, 2010–19: a systematic analysis for the Global Burden of Disease Study 2019. Lancet 2022; volume 400, issue 10352, 563-591.

2.     Cancer Incidence, Mortality, Years of Life Lost, Years Lived With Disability, and Disability-Adjusted Life Years for 29 Cancer Groups From 2010 to 2019. JAMA Oncology 2022; 8(3):420-444. doi:10.1001/jamaoncol.2021.6987

3.     The global burden of adolescent and young adult cancer in 2019: a systematic analysis for the Global Burden of Disease Study 2019. The Lancet Oncology 2022 Volume 23, Issue 1January 2022, Pages 27-52.

HPV and cancer data can be taken from a paper published in 2023.

https://www.mdpi.com/2076-393X/11/1/102

Figure 2: Quality of figure is not good. Also check/change the text font and writing style to improve the quality of figure.

Figure 3 is very basic and may be removed. You can add at least 1 more figure, with some important information.

Methodology section is totally missing. Why study was performed? What was the inclusion and exclusion criteria. How many studies were initially screened, studies excluded and included in the study.

Rewrite the conclusion of the paper. This is not the true conclusion of the paper. You have written about what you did in the paper. There are limited studies published on this issue and you recommend minimizing antibiotic resistance. What is coming out of your paper? Please write.

Overall paper has high scientific merit and can be published with incorporation of minor suggestions.

Author Response

Response to Reviewer’s Critiques

Reviewer 1:

The manuscript entitled “Bacterial infections and cancer: exploring the association and its implications for cancer patients” by Kafayat Yusuf et al., is well written and well presented. Authors highlights a very important issue of the association between bacterial infections and cancers. Paper included very important research work published on the role of different bacterial infections and how they aid in cancer progression.

We thank the reviewer for an insightful review of our manuscript.

Rewrite abstract of the paper and write key results in the abstract. The length of abstract is also short. Increase it to 300 words. Add the linkage of major bacterial infections with cancer in the abstract.

We have modified the abstract and increased the number of words as suggested.

The typical abstract is comprised of background, methods (search criteria, studies included etc), Results (for review it should be major part, with inclusion of key bacterial infections and their role in cancer), conclusion (2 sentences).

We have incorporated reviewer’s suggestion into the revised manuscript.

There are only a couple of papers published in 2022, which are included in the study. Add some latest data.

We have included the latest data as suggested.

Cite global cancer burden from the latest papers published in 2022 by Global Burden of Disease Research group, University of Washington, as pasted below.

  1. The global burden of cancer attributable to risk factors, 2010–19: a systematic analysis for the Global Burden of Disease Study 2019. Lancet 2022; volume 400, issue 10352, 563-591.
  2. Cancer Incidence, Mortality, Years of Life Lost, Years Lived With Disability, and Disability-Adjusted Life Years for 29 Cancer Groups From 2010 to 2019. JAMA Oncology 2022; 8(3):420-444. doi:10.1001/jamaoncol.2021.6987

3.The global burden of adolescent and young adult cancer in 2019: a systematic analysis for the Global Burden of Disease Study 2019. The Lancet Oncology 2022 Volume 23, Issue 1, January 2022, Pages 27-52.

 HPV and cancer data can be taken from a paper published in 2023.

https://www.mdpi.com/2076-393X/11/1/102

All three references have been included in the revised manuscript.

Figure 2: Quality of figure is not good. Also check/change the text font and writing style to improve the quality of figure.

We have improved the quality of Figure 2 in the revised manuscript.

Figure 3 is very basic and may be removed. You can add at least 1 more figure, with some important information.

In the revised manuscript, we have provided more details in Figure 3 about the complications that lead to increased bacterial infections and have also included names of the most common bacterial species that cause bacterial infections during cancer therapy.

Methodology section is totally missing. Why was study performed? What was the inclusion and exclusion criteria. How many studies were initially screened, studies excluded and included in the study.

We have now provided a Methods section as suggested by the reviewer.

Rewrite the conclusion of the paper. This is not the true conclusion of the paper. You have written about what you did in the paper. There are limited studies published on this issue and you recommend minimizing antibiotic resistance. What is coming out of your paper? Please write. Overall paper has high scientific merit and can be published with incorporation of minor suggestions.

In the revised manuscript, we have modified the Conclusions as suggested.  

Reviewer 2 Report

General comments:

The authors review the association of bacterial pathogens to cancer and the impact of bacterial infections in cancer patients. It is an overall interesting topic. The main issues I would like to address is the fact that the review bibliography is highly based in other reviews, and not in fundamental research articles. It would import to include the most recent research literature highlighting evidence of association of bacteria to cancer and its incidence, specific subtypes, and detail if the evidence of the % of chronically infected patients that develop cancer.

Also, would like to refer that part 4 (Bacterial infections during cancer treatment) should be improved in terms of text organization since it is dispersed. I would suggest including in figure 3 the type of infection caused by each species in cancer patients and ideally the frequency/incidence reported.

Minor comments/questions:

Overall article: Please review and uniformize the spaces before the reference number. In some sentences there is a space in others there is not. Please also review species name in italics.

Abstract

Line 15: suggestion: evade antibiotic/antimicrobial therapy

Section 2.1 Helicobacter pylori

What is the proportion of strains that secrete VacA? What is the proportion of H. pylori infected patients VacA positive that develop gastric cancer?

Section 2.3 Bacterial gastroenteritis

Salmonella: Any particular species or serovar?

Line 220:  full stop missing

Line 221: Please correct species name: pneumoniae

Line 227: regarding reference 66 I would suggest the authors refer the main conclusions of the epidemiological study

Line 238: obligate intracellular bacteria (not parasite). Please confirm the 50% rate of infections in adults. I would suggest the authors to search for a reference regarding epidemiology of Chlamydia pneumoniae in adult population.

Line 247: remove italics (During this time)

Line 294: Letter size seems different in this part of the text.

Section 4: Is there any epidemiological information regarding the incidence of each infection type during cancer treatment and what are the types of cancer that are more prone to bacterial infections ?

Section 5: I would suggest to use: antibiotics use and antimicrobial resistance.

Lines 358-361: I do not understand the meaning of this paragraph. Could you please reformulate?

Author Response

Comments and Suggestions for Authors

General comments:

The authors review the association of bacterial pathogens to cancer and the impact of bacterial infections in cancer patients. It is an overall interesting topic. The main issues I would like to address is the fact that the review bibliography is highly based in other reviews, and not in fundamental research articles. It would import to include the most recent research literature highlighting evidence of association of bacteria to cancer and its incidence, specific subtypes, and detail if the evidence of the % of chronically infected patients that develop cancer.

We thank the reviewer for the insightful comments and have revised the manuscript to incorporate the reviewer’s suggestions. 

Also, would like to refer that part 4 (Bacterial infections during cancer treatment) should be improved in terms of text organization since it is dispersed. I would suggest including in figure 3 the type of infection caused by each species in cancer patients and ideally the frequency/incidence reported.

We have improved part 4 as suggested. We have also modified Figure 3 as suggested by the reviewer. As far as details of the evidence of the % of chronically infected patients that develop cancer is concerned, there is no clear-cut data on the percentage/frequency of infections caused and as a result, we were unable to provide this information accurately.

Minor comments/questions:

Overall article: Please review and uniformize the spaces before the reference number. In some sentences there is a space in others there is not. Please also review species name in italics.

All the corrections have been made as suggested.

Abstract

Line 15: suggestion: evade antibiotic/antimicrobial therapy

Done as suggested

Section 2.1 Helicobacter pylori

What is the proportion of strains that secrete VacA? What is the proportion of H. pylori infected patients VacA positive that develop gastric cancer?

We provide this information in the revised manuscript.

Section 2.3 Bacterial gastroenteritis

Salmonella: Any particular species or serovar?

We provide this information in the revised manuscript.

Line 220:  full stop missing

Corrected

Line 221: Please correct species name: pneumoniae

Corrected

Line 227: regarding reference 66 I would suggest the authors refer the main conclusions of the epidemiological study

We have revised the manuscript as suggested by the reviewer.

Line 238: obligate intracellular bacteria (not parasite). Please confirm the 50% rate of infections in adults. I would suggest the authors to search for a reference regarding epidemiology of Chlamydia pneumoniae in adult population.

We provide the updated information in the revised manuscript.

Line 247: remove italics (During this time)

Corrected

Line 294: Letter size seems different in this part of the text.

Corrected

Section 4: Is there any epidemiological information regarding the incidence of each infection type during cancer treatment and what are the types of cancer that are more prone to bacterial infections?

We provide this information in the revised manuscript.

Section 5: I would suggest to use: antibiotics use and antimicrobial resistance.

Corrected

Lines 358-361: I do not understand the meaning of this paragraph. Could you please reformulate?

Simplified in the revision